# The Regulation of Flavivirus Infection by Hijacking Exosome-Mediated Cell–Cell Communication: New Insights on Virus–Host Interactions

**DOI:** 10.3390/v12070765

**Published:** 2020-07-16

**Authors:** José Manuel Reyes-Ruiz, Juan Fidel Osuna-Ramos, Luis Adrián De Jesús-González, Selvin Noé Palacios-Rápalo, Carlos Daniel Cordero-Rivera, Carlos Noe Farfan-Morales, Arianna Mahely Hurtado-Monzón, Carla Elizabeth Gallardo-Flores, Sofía L. Alcaraz-Estrada, Juan Santiago Salas-Benito, Rosa María del Ángel

**Affiliations:** 1Department of Infectomics and Molecular Pathogenesis, Center for Research and Advanced Studies (CINVESTAV-IPN), Mexico City 07320, Mexico; jmreyesr@cinvestav.mx (J.M.R.-R.); osram90@gmail.com (J.F.O.-R.); luis.dejesus@cinvestav.mx (L.A.D.J.-G.); selvin_palacios@yahoo.es (S.N.P.-R.); carlos.cordero@cinvestav.mx (C.D.C.-R.); carlos.farfan@cinvestav.mx (C.N.F.-M.); arianna.hurtado@cinvestav.mx (A.M.H.-M.); carla.gallardo@cinvestav.mx (C.E.G.-F.); 2División de Medicina Genómica, CMN 20 de Noviembre-ISSSTE, Mexico City 03229, Mexico; sofializeth@gmail.com; 3Maestría en Ciencias en Biomedicina Molecular, Escuela Nacional de Medicina y Homeopatía, Instituto Politécnico Nacional, Mexico City 07320, Mexico; 4Doctorado en Ciencias en Biotecnología, Escuela Nacional de Medicina y Homeopatía, Instituto Politécnico Nacional, Mexico City 07320, Mexico

**Keywords:** flavivirus, ZIKV, DENV, WNV, TBEV, exosomes, extracellular vesicles

## Abstract

The arthropod-borne flaviviruses are important human pathogens, and a deeper understanding of the virus–host cell interaction is required to identify cellular targets that can be used as therapeutic candidates. It is well reported that the flaviviruses hijack several cellular functions, such as exosome-mediated cell communication during infection, which is modulated by the delivery of the exosomal cargo of pro- or antiviral molecules to the receiving host cells. Therefore, to study the role of exosomes during flavivirus infections is essential, not only to understand its relevance in virus–host interaction, but also to identify molecular factors that may contribute to the development of new strategies to block these viral infections. This review explores the implications of exosomes in flavivirus dissemination and transmission from the vector to human host cells, as well as their involvement in the host immune response. The hypothesis about exosomes as a transplacental infection route of ZIKV and the paradox effect or the dual role of exosomes released during flavivirus infection are also discussed here. Although several studies have been performed in order to identify and characterize cellular and viral molecules released in exosomes, it is not clear how all of these components participate in viral pathogenesis. Further studies will determine the balance between protective and harmful exosomes secreted by flavivirus infected cells, the characteristics and components that distinguish them both, and how they could be a factor that determines the infection outcome.

## 1. Introduction

The flaviviruses, such as dengue (DENV), zika (ZIKV), tick-borne encephalitis (TBEV), and West Nile (WNV) viruses, which are transmitted by mosquitoes or ticks, cause a wide range of symptoms, such as paralysis, fever, meningitis, shock, congenital anomalies, and death [1,2,3,4,5]. Therefore, successful viral replication requires the hijacking of key cellular pathways within host cells [6,7]. One of these is cell-to-cell communication through the exosomal pathway, where exosomes released from human host and arthropod vector infected cells with flavivirus contain a broad set of viral and host cellular factors that can modify the recipient host cell responses [8,9,10,11,12,13,14,15,16].

Exosomes have a special place among the extracellular vesicles (EV) since they are different in size and origin [17] to other vesicles. Exosomes are 30 to 150 nm nanovesicles generated by the endosomal pathway of the intraluminal vesicles (ILVs) within multivesicular bodies (MBVs) [17,18,19]. In contrast, the other EVs, such as apoptotic bodies (ABs) and microvesicles, can measure up to 5000 nm and 1000 nm, respectively [20,21]. Additionally, in the case of ABs, the EV formation occurs during apoptotic disintegration, or the microvesicles bud directly from the plasma membrane [20,21]. It has been described that exosomes have a cargo that can consist of RNA, miRNA, proteins, or other molecules that modulate the cell communication, thereby triggering changes in the target cells during physiological mechanisms and disease as viral infections [22,23,24,25,26]. Moreover, exosomes are enriched in cholesterol [27,28] an essential molecule in flavivirus replication [29].

As a consequence, studies of exosomes released from the flavivirus-infected host and vector cells have shown highly specific populations with a molecular repertoire that determines their role in intracellular communication, viral dissemination, and host immune response [8,9,10,11,12,13,14,15,16,30]. Therefore, this review focuses on describing, comparing, and discussing the findings made so far to understand the role of exosomes during flavivirus infections.

## 2. Isolation Techniques and Heterogeneity in Exosomes Size

How should the characterization of exosomes produced from the flavivirus-infected host and vector cells be studied? It is difficult to answer this question due to the lack of standardization in exosome isolation and analysis methods [31]. The methods for exosome isolation include ultracentrifugation (UC) as the gold standard [32] and other techniques, such as Density Gradient (DG) and commercially available Exosome Precipitation Kits (EP Kits) [31]. Moreover, membrane proteins, such as tetraspanins CD63, CD81, and CD63 that are found on their surface, are used as specific markers of exosomes [33]. Thus, tetraspanins allow the characterization of heterogeneous populations of exosomes by immunoprecipitation (IP) [33]. Although IP could be considered as a way to purify exosomes, it is also a method to select a specific type of exosomes. Hence, it is possible that, under different isolation methods and cell lines, a different group of exosomes is enriched. This could explain the disparities in exosomal characteristics and compositions in studies that use different protocols to isolate exosomes. 

Currently, physical and compositional analyses are used for the characterization of isolated exosomes [31]. Within the physical analyses, there are those performed by transmission electron microscopy (TEM), Cryo-Electron Microscopy (Cryo-EM), Atomic Force Microscopy (AFM), Dynamic Light Scattering (DLS) and nanoparticle tracking analysis (NTA), all of which allow to determine the exosome size and/or concentration [31,34,35]. These strategies have given us valuable information about the size and shape, as well as the changes in both parameters under different conditions. Besides, techniques such as Western Blot, proteomic analysis, and next-generation sequencing (NGS) are used to study the composition of isolated exosomes [31,36,37].

The isolation and analysis methods currently used in the study of exosomes secreted from cells infected by flavivirus have shown a heterogeneous population of exosomes with different sizes, as shown in Table 1, and exosomal cargo, as shown in Figure 1. According to this, exosomes produced by flavivirus-infected arthropod vector cells are ~144 nm or ~30–250 nm in size when they are isolated by IP (positive to CD9 or CD63) [8,10] or DG [14,15], respectively, and analyzed with NTA, TEM, AFM, DLS, or Cryo-EM [8,10,14,15]. In effect, exosomes or EVs secreted from *Aedes albopictus* mosquito C6/36 cells infected with DENV had a size of 30–250 nm when isolated with the DG centrifugation technique [OptiPrep] and analyzed by Cryo-EM, as shown in Table 1 [14]. Additionally, exosomes from infected mosquito cells, positive to an ortholog of human CD9 (AalCD9) and isolated by IP, had sizes of ~80, ~95, and ~97 nm when they were analyzed by TEM, AFM, and DLS, respectively, as shown in Table 1 [10]. On the other hand, exosomes/EVs from flavivirus-infected human host cells were 30–250 nm in size if they were isolated by DG, UC, or EP Kit and analyzed with NTA, MET, or Cryo-EM, as shown in Table 1 [9,11,12,13,15,16]. This characterization suggests that exosomes from arthropod vector cells infected with flavivirus are larger than those released from uninfected arthropod vector cells [8,10], while exosomes released from uninfected or flavivirus-infected human cells are similar in size, as shown in Table 1 [9,11,12,13,15,16]. In this regard, it is possible that the heterogeneity of exosome populations could be related to their functions, exosomal cargo, and biogenesis. Furthermore, they have recently been classified into large (Exo-L, 90–120 nm) and small (Exo-S, 60–80 nm) exosome vesicles, based on their size [38]. 

Why are exosome size and/or cargo important? Exosome size and/or components located on their surface probably influence their recognition and capture by target cells [24]. After releasing, the recipient cells uptake those exosomes mainly through three different pathways: 1: endocytosis, 2: fusion, or 3: receptor–ligand interaction, any of the three is achieve by inducing internalization or eliciting intracellular signaling cascades [24,39,40,41,42,43]. It is known that exosomes released from flavivirus-activated platelets enhance the neutrophil extracellular trap (NET) formation and proinflammatory cytokine production through activation of the CLEC5A receptor on macrophages [12]. Additionally, exosome-mediated tick-borne flavivirus (Langat virus (LGTV), a pathogen closely related to TBEV) transmission to naïve cells, is receptor-dependent and requires clathrin-mediated endocytosis [15], whereas exosome-mediated mosquito-borne flavivirus (ZIKV) transmission is clathrin-independent [16].

Thus, the cargoes and sizes of exosomes from flavivirus-infected cells could affect their uptake by recipient cells and the modulation of cellular behavior between different hosts during infection, as described below.

## 3. Exosomes: A New Mechanism of Viral Dissemination Within and Among Hosts

The arthropod-borne flaviviruses (DENV, ZIKV, WNV and TBEV, among others) are replicated and assembled in close association with viral replication complexes (RC), where the NS4A and NS3 viral proteins play an essential role [44,45]. On the other hand, the ESCRT (endosomal sorting complex required for transport)-associated proteins and the Alix protein are involved in membrane deformation during viral particle morphogenesis [44,46]. Additionally, these cellular proteins are related to ILVs or exosome biogenesis within MVBs [47,48], which are found close to the RC and plasma membrane of flavivirus-infected cells [10]. The following will describe the state-of-the-art in how transmission routes through exosomes of different flavivirus infections occur.

### 3.1. DENV

Recently, our group reported the co-localization of Alix with prM/E viral proteins and the presence of virus-like particles in exosomes positive to CD9 of 90–100 nm (Exo-L) released from DENV-infected mosquito cells, as shown in Figure 1, and that they can infect naïve mosquito cells, as shown in Figure 2 [10]. This result was consistent with that previously reported by Vora et al., where they detected DENV E-protein in exosomes with size ranges of 50–100 nm (Exo-L), as shown in Figure 1 [14]. However, in this study, virus-like particles were not observed inside EVs positive to CD63 by immunoblotting, but the full-length DENV2 genome was found, as shown in Figure 1 [14]. Moreover, the Tsp29Fb tetraspanin, an ortholog of human CD63 identified in exosomes from DENV-infected mosquito cells, can interact with the E protein localized in the viral particle, as shown in Figure 1, to promote virus transmission in mouse monocyte-derived dendritic cells (Mo-DCs), human blood endothelial cells (HUVEC) and human skin keratinocytes (HaCaT cells), as shown in Figure 2 [14]. 

Additionally, it has recently been observed that exosomes from DENV-infected mosquito cells were positive to AalCD9, an ortholog of human CD81 (AalCD81), and Glyceraldehyde-3-Phosphate Dehydrogenase (GAPDH), as shown in Figure 1 [10]. In contrast, in EVs positives to CD63 by immunoblotting the NS1 protein and the fragment corresponding to the DENV C protein and HSP70 protein were found as shown in Figure 1 [14]. Accurately, in the case of DENV2, it has been described that viral infection activates human platelets via CLEC2 to release exosomes of 50–150 nm (Exo-L) that contains proteins, such as vinculin (VCL), calnexin (CANX), coagulation factor XIII A chain (F13A1), GAPDH, and tetraspanins CD9 and CD63, as shown in Figure 1 and Figure 4 [12]. Interestingly, Sung et al. concluded that DENV activates CLEC2 to release extracellular vesicles, and thereby activates NET formation and cytokine production in neutrophils and macrophages, as shown in Figure 4 [12]. Additionally, these EVs released from CLEC2-activated platelets contribute to increase permeability in HMEC-1 endothelial cells [12].

In this context, EVs from human monocyte-derived dendritic cells (mdDCs) infected with DENV3 contain the tetraspanins CD9, CD81, and CD63, and some viral components that can transfer the infection to C6/36 mosquito cells, as shown in Figure 4 [9]. These EVs from DENV-infected mdDCs also enclose a set of miRNAs, such as miR-4327, that could be a promising candidate for a circulating marker of severe dengue, as shown in Figure 4 [9], as well as mRNAs (DDX58, IFIT1, IFITM1) involved in DENV infection, as shown in Figure 5 [9]. Similarly, human alveolar basal epithelial (A549) cells infected with WNV release exosomes of ~100 nm (Exo-L) that enclose the miRNAs (miR-26a, miR-27a, and miR-29b) implicated in the virus infection, and some small noncoding RNAs (sncRNAs), such as U3, U97, and ACA31, as shown in Figure 1 and Figure 5 [11]. In the last few years, the induction of several miRNAs has been described in cells infected with different flaviviruses. Some of them are secreted via exosomes playing essential roles in cell-to-cell communication. Exosomes of ~100 nm (Exo-L), secreted from human adherent U937 macrophages infected with DENV, contain tetraspanins CD63 and CD81, microRNAs (miRNAs), such as miR-181a-5p, miR-4301, miR-4652-3p, intraluminal vesicle biogenesis-related proteins (Tsg101 and Alix), proteins with a role in the immune response (IL-6, C3, and Galectin-3), cytoskeleton proteins (Actin), enzymes (GAPDH, Enolase and AChE), fusion proteins (Annexins), and others proteins, such as Ubiquitin, HSC70, and H3, as shown in Figure 1 [13]. Interestingly, these exosomes contain the viral protease NS3 but do not possess the DENV polymerase NS5, as shown in Figure 1, and were not infectious to rhesus monkey kidney epithelial (LLC-MK2) cells [13] in contrast with exosomes released from flavivirus-infected arthropod vector cells [8,10,14,15]. It is possible, then, that at least some EVs from flavivirus-infected human host cells may not be involved in viral dissemination and they could induce changes in the host cell, since NS3 has recently been reported to alter the nuclear pore complex [49]. On the other hand, it has been observed that exosomes from DENV2-infected U937 macrophages keep inside some miRNAs, such as miR-181a-5p, miR-4301, and miR-4652-3p related to cell communication and cell adhesion [13]. The presence of these miRNAs could also be involved in viral pathogenesis.

Given the specific characteristics of exosomes and the presence of different proteins and RNAs, they can interact with the endothelial cells, inducing an alertness status in the endothelium and causing the activation and secretion of inflammatory mediators, such as TNF-α, INF-α, IL-6, IL-8, IL-10, IL-12p70, IP-10, and RANTES, establishing the first protective proinflammatory response during flavivirus infection [13].

### 3.2. ZIKV

A study conducted on C6/36 mosquito cells infected with ZIKV revealed that EVs positive to CD63 contain ZIKV E-protein on their surface, as well as phosphatidylserine (PS) and the ZIKV genome (E protein), as shown in Figure 1 [8]. In this context, Vora et al. suggest that E-protein is securely contained inside EVs; however, Martínez-Rojas et al. propose that it is found on the surface of EVs, as shown in Figure 1 [8,14]. Due to the pro-viral exosomal cargo of exosomes released from ZIKV-infected mosquito cells, they can infect human peripheral blood monocytes (THP-1 cells) and human endothelial vascular (HMEC-1) cells, as shown in Figure 2 [8].

Regarding the human host, EVs from primary culture cells of C57BL/6 mice cortical neurons, infected with ZIKV, contain tetraspanins CD9 and CD63, HSP70 protein, E viral protein, viral genome (NS5), and Sphingomyelinase 2 (nSMase2 or SMPD3), as shown in Figure 1, and they can infect primary cultures of murine cortical neurons (MCN), as shown in Figure 2 [16]. Therefore, exosomes released from ZIKV-infected cells promote viral spread.

### 3.3. Other Flaviviruses

Exosomes released from mouse brain microvascular endothelial cells (bEnd.3 cells), that constitute the blood-brain barrier and mouse neuroblastoma Neuro-2a cells (N2a cells), infected with LGTV contain viral RNA and viral proteins, as shown in Figure 1, and they can transmit the infection to neuronal cells similarly to exosomes produced from N2a cells infected with WNV, as shown in Figure 2 and Figure 3 [15]. In this sense, primary neuronal cultures of murine cortical neurons infected with LGTV generated exosomes that have a viral genome and viral proteins with the ability to infect the same kind of cells [15].

Moreover, exosomes from WNV-infected A549 cells contained HSP70, the tetraspanins CD9 and CD63 [11], and the mRNA, which encodes DDX58 [9,11], which plays an essential role in the host restriction of flavivirus infection, as shown in Figure 1 and Figure 5 [50,51]. Exosomes from N2a cells infected with LGTV contained HSP70 and CD9 cellular proteins, E and NS1 viral proteins, and both positive and negative sense LGTV RNA strands, as shown in Figure 1 and Figure 3, which were infectious for the naïve N2a cells [15]. Additionally, the viral genome was found in exosomes isolated from the WNV-infected N2a cells, as shown in Figure 3 [15], suggesting that exosomes produced during flavivirus infection can play a dual role since they contain molecules with pro-viral or antiviral activity. Also, it has been described that exosomes from DENV-infected arthropod cells, as well as exosomes from Ixodes scapularis ISE6 tick cells infected with LGTV, contain E and NS1 viral proteins, HSP70 cellular protein, and the positive and negative strands of LGTV RNA, as shown in Figure 1 [15], and they can infect human (HaCaT cells) and tick cells, as shown in Figure 3 [15]. Therefore, exosomes released from flavivirus-infected arthropod vector cells enclose the pro-viral exosomal cargo (viral proteins, virus genome, and viral particles), which is involved in the spread of the flavivirus to the human host or arthropod vector cells, as shown in Figure 2 and Figure 3 [8,10,14,15]. The release of the viral genome and/or viral particles inside exosomes can protect infective viral particles or the viral genome in such way that they can hide from immune surveillance, favoring viral transmission, as shown in Figure 2, Figure 3 and Figure 4.

Summarizing this section, the human host and arthropod vector cells infected with flavivirus produce distinct populations of exosomes with different sizes and exosomal cargo, which may have a pro- or antiviral role on recipient cells. Therefore, the selection of exosome sub-populations is of great interest for studies on exosome functions and biology and the development of exosome-based diagnostics and therapeutics.

## 4. The Role of Exosomes in the Host Immune Response During Flavivirus Infection

During flavivirus infection, a potent antiviral response is activated to prevent virus replication and spread to the neighboring cells in the human host [52,53,54]. This antiviral response is triggered by pattern recognition receptors (PRRs), such as the Toll-like receptors (TLRs), the nucleotide oligomerization domain (NOD)-like receptors (NLRs), and the retinoid-inducible gene I (RIG-I)-like receptors (RLRs), which recognize the flavivirus RNA and activate transcription factors IRF3, IRF7 and NF-kB via the signaling pathways for the production of type I interferons (IFN-α and β) and proinflammatory cytokines [53,55]. The IFN-α and β produced are released from the cell into the bloodstream and, once on the surface of target cells, bind to the IFN-α/β receptor (IFNAR) and activate the JAK/STAT signaling cascade, leading to the transcription of hundreds of IFN-stimulated genes (ISGs) with anti-flavivirus activity [54,56,57,58]. Thus, the cell-to-cell communication in response to flavivirus infection is mediated by the secretion of type I IFN [53]. However, exosomes can stimulate immune response as a new mechanism of cell signaling/communication during the different flavivirus infections. 

### 4.1. DENV

The IFN-inducible transmembrane protein 3 (IFITM3), a mediator required for the anti-flavivirus response of IFN [59,60,61], is found in exosomes derived from naïve HUVEC and human embryonic kidney 293 (293T) cells, which have antiviral activity in DENV2-infected HeLa cells, as shown in Figure 5 [30]. Additionally, Zhu et al. suggest that the IFITM-containing exosomes can be transferred to cells that are not directly stimulated with IFN, leading to the establishment of an antiviral state, as shown in Figure 5 [30]. 

The dendritic cells (DCs) are professional antigen-presenting cells (APCs), potent phagocytes [62], and critical biological targets of flavivirus infection [63]. In this regard, EVs released from mdDCs infected with DENV3 5532 (severe infection phenotype) contain miRNAs associated with DENV infection, such as let-7e, mir-1246, mir-1261, mir-142, mir-371b, mir-3937, and mir-4327, as shown in Figure 1 [9]. Interestingly, let-7e, mir-1261, mir-371b, and mir-4327 were found in EVs derived from mdDCs infected with the hemorrhagic DENV3-5532 strain, but not in cells infected with the mild DENV3-290 strain (isolated from a mild case of dengue). Thus, these miRNAs could be suitable circulating biomarkers of dengue disease severity, as shown in Figure 4 [9], as suggested by Mishra et al. [64]. The miRNAs are a type of small non-coding RNA that can regulate gene expression to modulate different biological processes [65]. Let-7e has pro-viral activity because it can inhibit TNF-α expression through its potential direct target to the enhancer of the Zeste homolog gene 2 (EZH2) in DENV2-infected human peripheral blood mononuclear cells (PBMCs) [66]. In contrast, some miRNAs found in EVs secreted by DENV3-infected mdDCs have antiviral activities, such as: the miR-1246 that regulates the NF-kB signaling to increase the proinflammatory response [67]; the mir-142 that can be inserted into the DENV2 genome to restrict the flavivirus replication in DCs and macrophages [68]; the expression of mir-3937 and mir-4327 could participate in response to flavivirus infection [69,70].

Martins et al. suggest that the miRNAs that were identified in EVs from mdDCs infected with DENV3-5532 regulate the transforming growth factor beta (TGF-β), Erb, MAP kinases (MAPK), phosphatidylinositol-3-kinase/Serine-threonine kinase (PIK3/AKT), and phosphatidylinositol pathways [9]. Hence, the interference of the PIK3/AKT and TGF-β pathways could facilitate DC apoptosis and a proinflammatory environment during DENV3-5532 infection [9,71]. Moreover, several miRNAs present in EVs released from mdDCs infected with DENV3-5532 can interfere with the mRNA surveillance pathway that degrades viral RNA, which may be associated with the severity of dengue [9,71]. In addition, mRNAs found in EVs derived from mdDCs infected with DENV3-5532 have been reported to be involved in the activation of the immune response pathways of T and B lymphocytes and DCs, as shown in Figure 5 [9]. On the other hand, mRNAs derived from EVs produced from mdDCs infected with DENV3-290 have an inhibitory effect on the immune response of DCs [9]. 

The interesting finding in the study of Martin et al. was that EVs, released by mdDCs infected with DENV3-5532, contain mRNAs of cytokines, such as CXCR4, macrophage migration inhibitory factor (MIF), IL-17A, and IL-8, involved in the disease severity in dengue patients and platelet and endothelial cell activation, and cytokines associated with plasma leakage and Dengue Shock Syndrome, such as IL-6, which coincides with the hemorrhagic manifestations in DENV3-5532 infected patients [9]. The CXCR4 could be considered a pro-viral factor since its antagonism increases T cell trafficking in the central nervous system, leading to a reduction in viral loads and a decrease in immunopathology at this site during the flavivirus encephalitis [72]. In this sense, the proinflammatory cytokine MIF is also a pro-viral factor because this molecule favors viral neuroinvasion by compromising the integrity of the blood-brain barrier in flavivirus infection [73]. Additionally, other mRNAs, such as ATP-dependent helicases (DDX58, DDX60, and DDX60L), chemokines (CXCL10 and CXCL11), and effectors of the type I IFN response (IFI35, IFI44L, IFIT1, IFIT5, IFIT3, and IFITM1) were found in EVs produced from mdDCs infected with DENV3-5532, as shown in Figure 1 and Figure 5 [9]. Therefore, EVs could be considered as valuable biomarkers to predict the clinical outcomes of flavivirus infection [9,71,74].

Furthermore, EVs isolated from PBMCs treated with IFN-α can inhibit DENV3-5532 infection in PBMCs, suggesting that EVs could be a communication pathway between immune cells to share defense signals during flavivirus infection to inhibit viral replication and reduce viral infection, as shown in Figure 1 and Figure 5 [9]. In this regard, Slonchak et al. suggest that flavivirus infection and IFN-α treatment predominantly stimulate the miRNAs, such as miR-27a, miR-26a, miR-29B, miR-3614, and miR-664a, and their incorporation into EVs, which can regulate the host genes involved in the immune response, as shown in Figure 1 and Figure 5 [11]. However, the flavivirus infection induces more profound changes in abundance of the sncRNAs and mRNAs present in EVs, in contrast with EVs derived from naïve cells treated with IFN [11]. Additionally, the mRNAs (IFNB1, IFNL2, RIG-I (DDX58), TRIM25, MDA5 (IFIH1), IRF1, IRF9, STAT2, ISG15, ISG54 (IFIT2), and OAS1) contained in EVs from the WNV-infected A549 cells or induced during flavivirus infection are associated with the antiviral response mediated by the type I IFN signaling pathway, Wnt signaling, Jak/STAT cascade, and T cell receptor signaling pathway [11]. In contrast, the mRNAs found in EVs released from IFN-treated cells only participate in the Wnt signaling and type I IFN signaling pathway [11]. Interestingly, the long (>300 nt) and small (<300 nt) RNAs incorporated into EVs from WNV-infected A549 cells, as shown in Figure 1, can induce the expression of genes, such as MDA5, TRIM25, and ISG15 involved in the innate immune response in naïve A549 cells [11]. Thus, exosomes/EVs secreted during flavivirus infection could contain proteins, miRNAs, mRNAs, and RNAs associated with antiviral response and transfer antiviral activity to the target cells, as shown in Figure 5.

DENV2 stimulates CLEC2 on human platelets to promote the release of EVs, which induces the NET formation, impairs endothelial integrity and aggravates the plasma leakage, contributing to DENV-induced mortality, as shown in Figure 4 [12]. Thus, EVs, CLEC2, and platelets play a critical role in the pathogenesis of flavivirus infections. Additionally, Sung et al. demonstrated that DENV2-activated platelets could release EVs to enhance NET production and the proinflammatory cytokine (TNF-α and IL-6) release via CLEC5A (DENV2 exosomes), as shown in Figure 4, and TLR2 (DENV2 microvesicles) in neutrophils and macrophages, supporting the argument that the platelet–neutrophil interactions contribute to an enhanced inflammatory reaction during DENV infection [12,75]. Hence, Sung et al. suggest that DENV can activate CLEC2 to trigger the NALP3 inflammasome and induce IL-1β release from platelets to improve proinflammatory cytokine release and NET formation via CLEC5A and TLR2 [12]. Thus, NET formation induced by DENV could increase the platelet–neutrophil interaction via the release of histone 2A and other nuclear components to activate the platelets [12]. However, the amount of IL-1β produced by the platelets is low compared to that produced by macrophages, since high levels of IL-1β, IL-18, and the activation of the NLRP3 inflammasome have been reported from GM-Mφ (inflammatory macrophages), through the activation of caspase-1. Additionally, it has been found that CLEC5A is essential for the induction of NLRP3 in GM-Mφ during DENV infection and the activation of TH17 lymphocytes, which can contribute to the host’s immune response against the virus [76,77]. As a result of that, the activation of CLEC2 and CLEC5A/TLR2 in the platelets and leukocytes by EVs contributes to dengue disease severity and they could be a potential target for inhibiting flavivirus infections [12].

On the other hand, EVs released from DENV2-infected U937 macrophages are also able to modulate the endothelial response in EA.hy926 cells [13] because they increase the expression of adhesion molecules, such as vascular endothelial-cadherin (VE-Cad; major molecule that controls cellular junctions and blood vessel formation) and intercellular adhesion molecule 1 (ICAM-1; endothelial molecule involved in the inflammation and the regulation of vascular permeability) [13], and the activation and secretion of inflammatory mediators, such as IL-8, INF-α, IL-12p70, TNF-α, IL-10, and IP-10, in contrast with the cells treated with EVs from naïve U937 cells [13]. These EVs also contain miRNAs (15–25 nt): miR-4652-3p, miR-4301, and miR-181a-5p with gene targets related to cell adhesion and cell communication regulation [13], suggesting that these miRNA have a role in endothelial activation and could regulate the genes implicated in the barrier and structural functions of vessels in flavivirus infection [13]. Specifically, the miR-4301 has an antiproliferative and pro-apoptotic role in cancer [78,79,80], whereas miR-181a-5p negatively regulates the inflammatory response [81,82,83] that is involved in endothelial activation during DENV infection [13]. In contrast, some cellular proteins involved in the immune response, such as C3 complement protein, metalloproteinase inhibitor (MPI), and galectin-3 (Gal-3), were found inside EVs secreted by DENV-infected macrophages, as shown in Figure 5 [13]. 

### 4.2. ZIKV

Regarding ZIKV, EVs from ZIKV-infected mosquito cells can induce the differentiation and expression of CD11b+ on the surface of monocytes [8]. Additionally, the naïve THP-1 cells activated by EVs released from infected mosquito cells show a change of naïve cells to an adherent phenotype, as occurs during ZIKV infection [8]. Thus, EVs from ZIKV-infected C6/36 cells can activate and differentiate the naïve monocytes. 

Moreover, EVs from ZIKV-infected mosquito cells can induce TNF-α mRNA expression in naïve THP-1 and HMEC-1 cells, which is a host factor involved in nervous system damage caused by ZIKV infection [8]. The authors suggest that the induction of TNF-α mRNA can induce changes in the proinflammatory phenotype, playing a role in the immune response [8].

Additionally, EVs from ZIKV-infected C6/36 cells induce high levels of tissue factor (TF) and the activation of protease-activated receptor-1 (PAR-1) in naïve HMEC-1 cells [8]. TF is an essential factor in the blood coagulation mechanism since it induces the production of hemostatic proteases (thrombin) to activate PAR-1, which is involved in the expression of pro-adherent and proinflammatory molecules [8]. Therefore, Martínez-Rojas et al. suggest that EVs from ZIKV-infected cells contribute to pathogenesis during ZIKV infection [8], since these EVs can infect endothelial vascular cells with proinflammatory cytokine expression to increase vascular permeability and endothelial damage [8].

Thus, exosomes/EVs secreted from flavivirus-infected cells can stimulate the endothelium and other immune cells to modulate the cell environment, promoting an immune response or contributing to the pathogenesis of flavivirus disease.

## 5. Exosomes as a Transplacental Infection Route of ZIKV: A Hypothesis

The human placenta is a complex organ for maternal–fetus metabolic exchange [84,85,86]. Although the placenta serves as a protective barrier, some infectious agents can pass through and colonize this organ [87,88,89], either via the maternal blood or by ascending the genital tract [87].

Therefore, despite the limitation that the placenta represents in the transmission of pathogens to the fetus, it has been reported that ZIKV can infect placental cells, such as trophoblasts and fetal macrophages, as well as cells composing the decidual basalis [90,91,92]. Besides, previous studies have discovered the presence of ZIKV RNA in the amniotic fluid of fetuses, as well as human and mouse trophoblasts [93,94,95]. Furthermore, 42 to 54 nm virus particles have been found by negative staining in fetal brain tissue [96], which confirms that ZIKV can cross the placenta. One possible explaination of these findings is the fact that viral infection induces damage on the syncytiotrophoblast, and it may facilitate the entry of ZIKV into the cytotrophoblast and other fetal tissues [93,97]. Another explanation could be that ZIKV transmission occurs through exosomes.

Autophagy is a mechanism that the placenta uses for its defense against pathogens [98]. However, some pathogens can hijack this mechanism and use it as a mean of survival and propagation, as may be occurring during ZIKV infection, because this flavivirus induces autophagy [99,100,101]. Thus, the machinery of autophagy promotes exosome biogenesis; in this sense, autophagy and exosome biogenesis are two related processes [102,103,104]. 

Therefore, the possibility of placental colonization by ZIKV and subsequent fetal infection could occur by secretory autophagy [105,106,107] and through exosomes, since flavivirus exploits this pathway to spread infection [10,14,15,16]. However, the transmission of ZIKV through exosomes by the placenta has not been reported so far.

## 6. Concluding Remarks

Given the nature and importance of exosomes in the cell life cycle, they constitute an excellent option for cell-to-cell communication, mainly after infection with different viruses. Exosomes can act as a protective mechanism against infection, but they can also contribute or favor viral infection and pathogenesis. During flavivirus infection, different cellular and viral molecules are released inside exosomes and EVs by different types of infected cells. Although several studies have been performed in order to identify and characterize the mRNA, proteins, miRNA, and other components present in exosomes and EVs, it is not yet clear how all these components participate in viral pathogenesis. Further studies of the viral and cellular components released in EV and exosomes will help us to use them in drug development or in the modulation of infection severity.

It is possible that exosomes or EVs that favor viral infection and exosomes/Evs involved in viral infection inhibition can coexist in vivo in a dynamic balance between protective exosomes and flavivirus-manipulated exosomes. The balance between both types of exosomes/Evs could dictate the infection outcome.

## Figures and Tables

**Figure 1 viruses-12-00765-f001:**
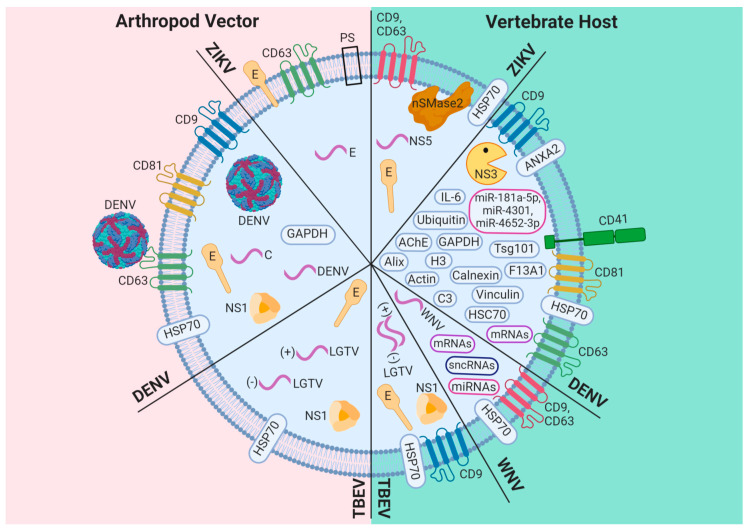
**Composition of exosomes/EVs released from arthropod vector and vertebrate host cells infected with flavivirus.** Exosomes/EVs produced by flavivirus-infected cells contain a wide variety of cellular components, including sncRNAs, miRNAs, mRNAs, exosome markers (Alix, Tsg101, HSP70, CD9, CD81, CD63, GAPDH, Annexin A2 (ANXA2)), proteins involved in the immune response (IL-6 and complement component 3 (C3)), enzymes (acetylcholinesterase (AChE) and neutral sphingomyelinase 2 (nSMase2)), lipids, such as phosphatidylserine (PS), and others proteins, such as ubiquitin, Histone 3 (H3), calnexin, coagulation factor XIII A chain (F13A1), vinculin, actin, CD41, and HSC70. Additionally, they can enclose viral components, such as proteins (E and NS1), viral RNA (pink curve line) strands with positive (+) or negative (−) sense, and viral particles.

**Figure 2 viruses-12-00765-f002:**
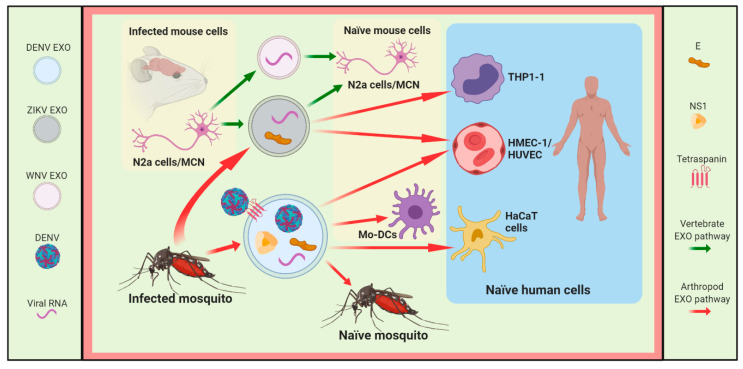
**The role of exosomes/EVs during DENV, ZIKV, and WNV infections.** Exosomes (EXO)/EVs secreted from mosquito and vertebrate cells infected with flavivirus contain viral components to infect the naïve human, vertebrate, or mosquito cells. Abbreviations: N2a cells, mouse neuroblastoma Neuro-2a; MCN, primary cultures of murine cortical neurons; Mo-DCs, mouse monocyte-derived dendritic cells; THP-1, human monocytes from peripherical blood; HMEC-1, human endothelial cells; HUVEC, human umbilical vein endothelial cells; HaCaT cells, human-skin keratinocytes.

**Figure 3 viruses-12-00765-f003:**
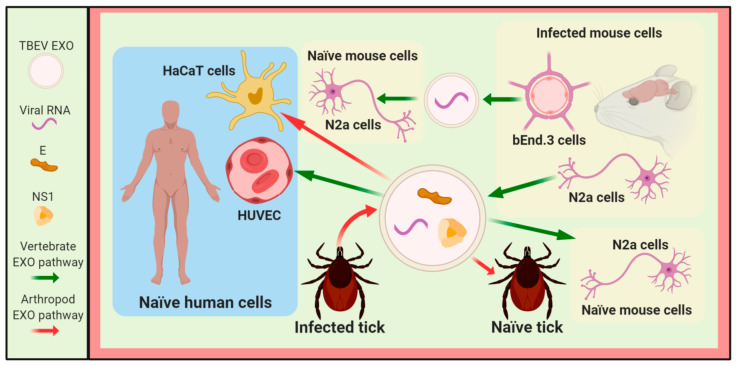
**Exosomes/EVs derived from TBEV infection promote viral dissemination.** Exosomes (EXO)/EVs secreted from vertebrate and tick cells infected with TBEV contain viral components to infect the naïve human, vertebrate, or tick cells. Abbreviations: N2a cells, mouse neuroblastoma Neuro-2a; HUVEC, human umbilical vein endothelial cells; HaCaT cells, human skin keratinocytes; bEnd.3 cells, mouse brain-microvascular endothelial cells that constitute the blood-brain barrier.

**Figure 4 viruses-12-00765-f004:**
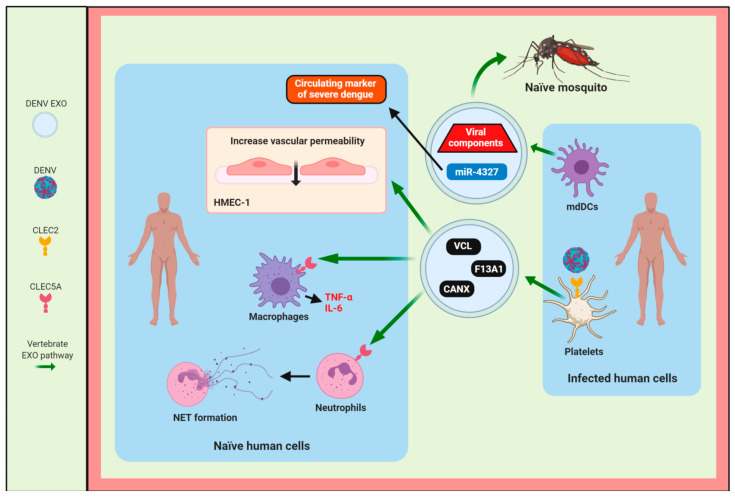
**Exosomes/EVs are involved in the viral dissemination and pathogenesis of DENV infection.** EXO/EVs produced by flavivirus-infected human cells enclose miRNAs, such as miR-4327 (candidate for circulating marker of severe dengue), cellular proteins, and viral components that promote infection to naïve mosquito and human cells. Additionally, they can increase vascular permeability; enhance the neutrophil extracellular trap (NET) formation and proinflammatory cytokine (TNF-α and IL-6) production through activation of the CLEC5A receptor on macrophages. Abbreviations: HMEC-1, human endothelial cells; HUVEC, human umbilical vein endothelial cells; mdDCs, human monocyte-derived dendritic cells.

**Figure 5 viruses-12-00765-f005:**
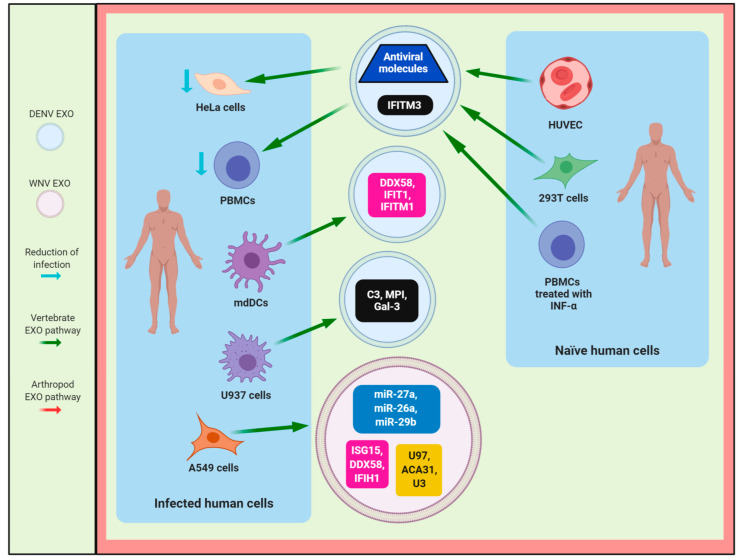
**The protective role of exosomes/EVs during flavivirus infection.** Exosomes/EVs from flavivirus-infected, naïve, or treated with INF-α human cells containing sncRNAs (yellow box), miRNAs (blue box), mRNAs (pink box) and proteins (black box), which are involved in reducing infection in human cells. Abbreviations: HUVEC, human umbilical vein endothelial cells; mdDCs, human monocyte-derived dendritic cells; HeLa cells, cervical cancer cells; PBMCs, human peripheral blood mononuclear cells; U937 cells, human adherent U937 macrophages; A549 cells, human alveolar carcinoma cells; 293T cells, human embryonic kidney cells.

**Table 1 viruses-12-00765-t001:** The size of exosomes/EVs secreted by flavivirus-infected human host and arthropod vector cells.

Flavivirus	Exosomes				Size (nm)		Reference
	Isolation Methods	Analysis Methods	Arthropod Vector	Vertebrate Host	Uninfected	Infected	
ZIKV	IP (CD63 positive)	NTA	C6/36 cells		268.9 ± 8.20 107.8 ± 3.10	319.3 ± 11.50 125.5 ± 1.60	[8]
ZIKV	DG	Cryo-EM		Primary culture of C57BL/6 mice cortical neurons	30-350 [50-200 (↑), 200-350 (↓)]	30-350 [50-150 (↑), 150-350 (↓)]	[16]
DENV2	IP (CD9 positive)	TEM	C6/36 cells		46.77 ± 2.98	81.18 ± 4.70	[10]
DENV2	IP (CD9 positive)	AFM	C6/36 cells		55.91 ± 2.71	95.07 ± 13.34	[10]
DENV2	IP (CD9 positive)	DLS	C6/36 cells		42.77 ± 2.29	97.19 ± 10.50	[10]
DENV2	DG	Cryo-EM	C6/36 cells		30-250 [50-100 (↑), 150-200 (↓)]	30-250 [50-100 (↑), 100-150 (↓)]	[14]
DENV2	UC	MET		U937 cells	N/A	~100	[13]
DENV2	UC	NTA and TEM		Human Platelets	50-150	50-150	[12]
DENV3	UC	MET		mdDCs	30-180	30-180	[9]
DENV3	UC	NTA		mdDCs	<200 [100 (↑)]	<200 [100 (↑)]	[9]
WNV	EP Kit	TEM		Lung cancer A549 cells	~100	~100	[11]
LGTV (TBEV)	DG	Cryo-EM	*Ixodes scapularis* ISEG tick cells		30-250 [50-100 (↑), 150-250 (↓)]	30-250 [50-100 (↑), 100-250 (↓)]	[15]
LGTV (TBEV)	DG	Cryo-EM		N2a neuronal cells	30-250 [50-100 (↑), 150-250 (↓)]	30-250 [50-100 (↑), 150-250 (↓)]	[15]
LGTV (TBEV)	EP Kit	Cryo-EM		N2a neuronal cells	30-200	30-200	[15]

Abbreviations: IP, Immunoprecipitation; DG, Density Gradient; UC, Ultracentrifugation; EP Kit, commercially available Exosomes Precipitation Kits; NTA, Nanoparticle Tracking Analysis; Cryo-EM, Cryo-Electron Microscopy; TEM, Transmission Electron Microscopy; AFM, Atomic Force Microcopy; DLS, Dynamic Light Scattering; (↑), Many EVs that size; (↓), Less EVs that size; mdDCs, Human monocyte-derived dendritic cells.

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
