# Peer review of "The Regulation of Flavivirus Infection by Hijacking Exosome-Mediated Cell–Cell Communication: New Insights on Virus–Host Interactions"

_viruses, 2020, doi:10.3390/v12070765_

Round 1

Reviewer 1 Report

This manuscript gives a comprehensive review regarding the composition, content, and functions of exosomes release from various cell types during flavivirus infection. This topic is very important because dengue virus is the most prevalent vector-borne diseases, but specific treatment and effect vaccines are still not available. Thus, flavivirus-induced exosomes and exosomes-induced inflammatory reactions would shed light to develop novel strategy for the development of effective vaccine and therapeutic agents. However, minor corrections are necessary to it is accepted for publication.

  • LINE 42: What does ‘ABs’ mean? The author should give a full name and give a brief description.
  • Table 1: DENV2-induced EVs from activate platelets are examined not only by NTA, but also by TEM.
  • Figure 2: The content is too complicated, and I suggest to remove the results from cell lines, and only use data from DV-infected primary cells. We already understand the cytokines and several immune responses from primary macrophages are distinct from cell lines.
  • Line 319: The description is incorrect. What Sung says is that DENC activate CLEC2 to release extracellular vesicles, thereby activate NET formation and cytokine production in neutrophils and macrophages.

DENV does activate NALP3 inflammasome dramatically in inflammatory macrophages

  • BLOOD: https://ashpublications.org/blood/article/121/1/95/31093/CLEC5A-is-critical-for-dengue-virus-induced
  • J Biomed Sci: https://www.ncbi.nlm.nih.gov/pmc/articles/PMC3686598/

Compared to macrophages, the amount of IL-1beta from activated platelets are very low. The author should cite these two papers and clarify this point. Macrophage is the major source of IL-1 beta. The author should read the original papers and compare the IL-1 amount to clarify this point. I recommend to add these two papers to the reference.

 PS: My affiliations is not in Yang-Ming University now. Please change to

Genomics Research Center, Academia Sinica, Taipei, Taiwan  

Author Response

This manuscript gives a comprehensive review regarding the composition, content, and functions of exosomes release from various cell types during flavivirus infection. This topic is very important because dengue virus is the most prevalent vector-borne diseases, but specific treatment and effect vaccines are still not available. Thus, flavivirus-induced exosomes and exosomes-induced inflammatory reactions would shed light to develop novel strategy for the development of effective vaccine and therapeutic agents. However, minor corrections are necessary to it is accepted for publication.

1) LINE 42: What does ‘ABs’ mean? The author should give a full name and give a brief description.

Reply: The meaning of "ABs" is given in the first paragraph of the introduction (line 67), and in the same line we explained that this type of extracellular vesicles are apoptotic bodies with a size of 5000 nm. Citations 20 and 21included in the reference list provide more information about this type of vesicles.

2) Table 1: DENV2-induced EVs from activate platelets are examined not only by NTA, but also by TEM.

Reply: In the revised version of the manuscript, the TEM analysis of DENV2-induced EVs was included in the Table 1.

3) Figure 2: The content is too complicated, and I suggest to remove the results from cell lines, and only use data from DV-infected primary cells. We already understand the cytokines and several immune responses from primary macrophages are distinct from cell lines.

Reply: In the new version of the manuscript, the information included in Figure 2 was modified to make it easier to understand and now is included in Figures 2, 3, 4 and 5. Also, we decided to keep the results of the flavivirus-infected secondary cell lines in this figure because they could help to demonstrate the virus-host interaction.

4) Line 319: The description is incorrect. What Sung says is that DENC activate CLEC2 to release extracellular vesicles, thereby activate NET formation and cytokine production in neutrophils and macrophages.

Reply: The sentence: “DENV2 stimulates CLEC2 on human platelets to promote the release of EVs, leading to the NET formation…” was changed to “Sung et al. concluded that DENV activate CLEC2 to release extracellular vesicles, thereby activate NET formation and cytokine production in neutrophils and macrophages” (Line 184) as suggested the reviewer.

Moreover, the sentence: “DENV2 stimulates CLEC2 on human platelets to promote the release of EVs, leading to the NET formation..” was changed to “DENV2 stimulates CLEC2 on human platelets to promote the release of EVs which induce the NET formation…” in this new submission as suggested the reviewer (Line 382).

5) DENV does activate NALP3 inflammasome dramatically in inflammatory macrophages.

BLOOD: https://ashpublications.org/blood/article/121/1/95/31093/CLEC5A-is-critical-for-dengue-virus-induced

J Biomed Sci: https://www.ncbi.nlm.nih.gov/pmc/articles/PMC3686598/

Compared to macrophages, the amount of IL-1beta from activated platelets are very low. The author should cite these two papers and clarify this point. Macrophage is the major source of IL-1 beta. The author should read the original papers and compare the IL-1 amount to clarify this point. I recommend to add these two papers to the reference.

PS: My affiliations is not in Yang-Ming University now. Please change to

Genomics Research Center, Academia Sinica, Taipei, Taiwan

Reply: The information on the amount of IL-1β produced during dengue virus infection and references were added as suggested by the reviewer as follows:

“However, the amount of IL-1β produced by platelets is low compared to that produced by macrophages, since high levels of IL-1β, IL-18, and activation of NLRP3 inflammasome have been reported from GM-Mφ (inflammatory macrophages), through activation of caspase-1. Also, it has been found that CLEC5A is essential for the induction of NLRP3 in GM-Mφ during DENV infection and the activation of TH17 lymphocytes which can contribute to the host's immune response against the virus [71,72]” (Text and references were added in the line 396).

Reviewer 2 Report

Reyes-Ruiz et al., reviewed exosomes mediated cell communication in all flavivirus infections in an elaborate manner. Cartoons describing in Figures 1 and 2 are really impressive.   

As a reader, I felt it is a more informative and excellent resource for other flavivirus researchers.  But one quick comment would be if the authors provide the subheadings on 2. Exosomes heterogeneity, 3. new transmission route of flavivirus infection, and 4. the role of exosomes in the host immune response.  As the authors were concentrating on ZIKV, DENV2, 3, WNV ad TBEV, they can easily make subheadings, which would make the reader read and understand better.  Because of that in some places, it looks too much information and the reader will loss of train of thoughts.  

Figure 2 can be divided into two or three cartoons if possible for the benefit of readers. 

Again the authors have referred almost 100 research articles, so putting little effort would definitely enhance the quality of the article.

Author Response

1) As a reader, I felt it is a more informative and excellent resource for other flavivirus researchers.  But one quick comment would be if the authors provide the subheadings on 2. Exosomes heterogeneity, 3. new transmission route of flavivirus infection, and 4. the role of exosomes in the host immune response.  As the authors were concentrating on ZIKV, DENV2, 3, WNV ad TBEV, they can easily make subheadings, which would make the reader read and understand better.  Because of that in some places, it looks too much information and the reader will loss of train of thoughts.

Reply: The subheadings for each flavivirus (1.DENV, 2.ZIKV and 3. other flaviviruses, included WNV and TBEV) were provided as subsections in the section 3. new route of transmission of flavivirus infection, and 4. the role of exosomes in the host immune response. However, the section 2. Exosome heterogeneity, was described in a general way, not specifically for each virus.

2) Figure 2 can be divided into two or three cartoons if possible for the benefit of readers.

Reply: In the revised version of the present manuscript the Figure 2 was divided into five cartoons to make it easier to understand (Figures 2, 3, 4 and 5).

3) Again the authors have referred almost 100 research articles, so putting little effort would definitely enhance the quality of the article.

Reply: We believe that the references included here are necessary to achieve the objective of this review.

Reviewer 3 Report

Review viruses-791255

This review summarizes the published data in the relatively new field of Exosome vesicles and flaviviral infections. This is a subject of interest to most flavivirologists, however the review written as it is, needs some major changes to get the key messages across.

General comments:

  • The authors do not package the literature reviewed in the field of exosomes and flaviviral infections as a story with insightful conclusions. Currently, the review reads like a list of facts with no real synthesis of the information.
  • Furthermore, the authors do not highlight the gaps of knowledge in the field and the possible inconsistencies between published data.
    • Additionally, it would be great if the authors can suggest ways to go about filling those knowledge gaps, such as experiment and studies that could be conducted.
  • The authors do not discuss whether there are any epidemiological or clinical data to suggest that the formation of exosomes is clinically relevant for either spread of infection or severe disease. If such data is missing, then it’s good to mention that.
  • At the end of this review, it is unclear whether exosomes are pro-viral or anti-viral, or both? If currently, the known information is insufficient to make a judgement either way, then please do mention that. At the moment, it is unclear to the reader.
  • There are a lot of grammatical, and spelling mistakes. This manuscript needs to be proof read to correct those.
  • Overuse and sometime improper use of the phrase “In this sense,..”
  • Line 46 –Acronym “ABs” needs to be defined

Abstract

  • Good to have a sentence with the author’s conclusion at the end of the abstract.

Section #2,

  • The subheading is misleading – do you mean that there is heterogeneity in the techniques used, or in the actual exosome size and components, or Both? This needs to be clear in both the subheading and the text.
  • Discuss the exosomal size of different Flaviruses… does published data show that different flaviviral infections lead to different exosomal sizes?
  • Table 1. Presents various studies that have also used various different mammalian cell lines. Can the authors discuss whether the heterogeneity in exosomal sizes can also be due to the different cell lines used in those studies?

Section #3

  • The authors should not describe the spread of virus through exosomes as a “new route of transmission”. Transmission route between human to human is through mosquitos. Rephrase the section#3 subheading to something along the lines of a new mechanisms of viral dissemination within hosts.
  • Is it known how these exosomes can enter other cells? Please discuss.

Author Response

This review summarizes the published data in the relatively new field of Exosome vesicles and flaviviral infections. This is a subject of interest to most flavivirologists, however the review written as it is, needs some major changes to get the key messages across.

General comments:

The authors do not package the literature reviewed in the field of exosomes and flaviviral infections as a story with insightful conclusions. Currently, the review reads like a list of facts with no real synthesis of the information.

Furthermore, the authors do not highlight the gaps of knowledge in the field and the possible inconsistencies between published data.

Additionally, it would be great if the authors can suggest ways to go about filling those knowledge gaps, such as experiment and studies that could be conducted.

The authors do not discuss whether there are any epidemiological or clinical data to suggest that the formation of exosomes is clinically relevant for either spread of infection or severe disease. If such data is missing, then it’s good to mention that.

At the end of this review, it is unclear whether exosomes are pro-viral or anti-viral, or both? If currently, the known information is insufficient to make a judgement either way, then please do mention that. At the moment, it is unclear to the reader.

Reply: Considering the reviewer's observations, we carried out a restructuring of the manuscript, we provided few more details for each virus, we also added partial conclusions, discussing findings and proposing future elements to study.

  1. There are a lot of grammatical, and spelling mistakes. This manuscript needs to be proof read to correct those.

Reply: The grammar of the manuscript was checked and corrected.

  1. Overuse and sometime improper use of the phrase “In this sense,..”

Reply: The use of the phrase “In this sense…” was modified or removed in this new submission. Changes made:

 “In this sense”, present in line 76, was changed with the sentence “As a consequence,”

“In this sense”, present in line 110, was changed with the sentence “According to this,”

“In this sense”, present in line 136, was changed with the sentence “It is known that,”

“In this sense”, present in line 249, remains unchanged

“In this sense”, present in line 230, was changed with the sentence “In this context,”

 “In this sense”, present in líne 351, remains unchanged

“In this sense”, present in líne 473, was changed with the sentence “As a result of that,”

  1. Line 46 –Acronym “ABs” needs to be defined.

Reply: The meaning of "ABs" was included in the first paragraph on line 67,

Abstract

  1. Good to have a sentence with the author’s conclusion at the end of the abstract.

Reply: The abstract was modified as suggested by the reviewer.

At the end of the Abstract: “Although several studies have been performed trying to identify and characterize cellular and viral molecules released in exosomes, it is not clear how all these components participate in viral pathogenesis. Further studies will determine the balance between protective and harmful flavivirus exosomes, the characteristics and components that distinguish between both, and how they could be a factor that determines the infection outcome.

Section #2,

  1. The subheading is misleading – do you mean that there is heterogeneity in the techniques used, or in the actual exosome size and components, or Both? This needs to be clear in both the subheading and the text.

Reply: The sentence: “2. Exosomes Heterogeneity: Isolation Techniques and Subpopulations” was changed to “2. Isolation Techniques and Heterogeneity in Size of Exosomes” in this new submission.

  1. Discuss the exosomal size of different Flaviruses… does published data show that different flaviviral infections lead to different exosomal sizes?

Table 1. Presents various studies that have also used various different mammalian cell lines. Can the authors discuss whether the heterogeneity in exosomal sizes can also be due to the different cell lines used in those studies?

Reply: This possibility was discussed in the new version of the manuscript as suggested by the reviewer (Line 93).

“Although the IP could be considered as a very specific way to purify exososomes, it is also a method to select a specific type of exosomes. Thus, it is possible that under different isolation methods and cell lines, different group of exosomes are enriched. It could explain the differences in exosomal characteristics and composition in studies which use different protocols to isolate exosomes.”

Section #3

  1. The authors should not describe the spread of virus through exosomes as a “new route of transmission”. Transmission route between human to human is through mosquitos. Rephrase the section#3 subheading to something along the lines of a new mechanisms of viral dissemination within hosts.

Reply: The sentence: “3. Exosomes: A New Transmission Route of Flavivirus Infection” was changed to “3. Exosomes: A New Mechanism of Viral Dissemination Within and Among Hosts”.

  1. Is it known how these exosomes can enter other cells? Please discuss.

Reply: The answer to this question is discussed in this new submission:

“Exosome size and/or components located on their surface probably influence their recognition and capture by target cells [24]. After releasing, the recipient cells mainly uptake those exosomes through three different pathways: 1: endocytosis, 2: fusion or 3: receptor-ligand interaction, either by inducing internalization or eliciting intracellular signaling cascades [24,36–40]. In this sense, exosomes released from flavivirus activated platelets enhance the neutrophil extracellular traps (NET) formation and proinflammatory cytokine production through activation of the CLEC5A receptor on macrophages [12]. Also, exosome mediated tick-borne flavivirus (Langat virus (LGTV), a pathogen closely related to TBEV) transmission to naïve cells, is receptor-dependent and requires clathrin-mediated endocytosis [15], whereas exosome-mediated mosquito-borne flavivirus (ZIKV) transmission is clathrin-independent [16].”

Reviewer 4 Report

                This manuscript represents a detailed overview of the available literature documenting exosome associations with flavivirus infections. I have one major recommendation regarding the review:  while the manuscript contains a lot of information, the organization and flow of the paragraphs tends to camouflage the key points that are being made.  This makes it difficult for the reader to digest the information and the take home message gets lost.  In editing/re-organizing the manuscript, the authors may wish to also note that Mishra et al published a related review on the topic in 2019 (PMID 31711408). Finally, I would also recommend that the authors review the entire manuscript to optimize the use of the English language.

Some specific points/suggestions:

  1. The title should be rewritten to better accommodate standard usage of the English language.
  2. All acronyms must be clearly defined for the reader the first time they appear. For example, on line 46, what does AB refer to?
  3. Line 48: fix:  acceptor o target
  4. Line 57: fix: an essential factor to considerer
  5. Line 68-70: as written the sentence that starts ‘While….’ Is not a complete sentence.  Please fix
  6. Line 66: Cry-EM or Cryo EM as is used later in the MS
  7. 2: I find this figure to be very complex and difficult to understand given the explanation in the legend.  My interpretation is that the authors are trying to include too much information into the graphic, making it difficult for the reader to digest what the take home points are.  Thus I do not find this figure to be particularly useful and would recommend that it be significantly re-drawn.
  8. Line 184 – add the unit to the number in ….~100 (Exo-L) secreted…
  9. Section 3: This section is difficult for the non-expert reader to keep straight due to the inclusion of many details that are not put into proper perspective or flow.  Thus while it provides an overview of the literature in the area, it does not do it in a very approachable fashion – which I fear will significantly limit the impact of the review.
  10. Line 326: the statement….EVs from ZIKV-infected mosquito cells can differentiate… implies that EVs differentiate – please reword for clarity
  11. Line 362: A hypothesis, not ‘an’ hypothesis

Author Response

1) This manuscript represents a detailed overview of the available literature documenting exosome associations with flavivirus infections. I have one major recommendation regarding the review:  while the manuscript contains a lot of information, the organization and flow of the paragraphs tends to camouflage the key points that are being made.  This makes it difficult for the reader to digest the information and the take home message gets lost.  In editing/re-organizing the manuscript, the authors may wish to also note that Mishra et al published a related review on the topic in 2019 (PMID 31711408). Finally, I would also recommend that the authors review the entire manuscript to optimize the use of the English language

Reply: The corrections suggested by the reviewer were performed in this new submission. Also, Mishra et al article was included in the

Some specific points/suggestions:

2) The title should be rewritten to better accommodate standard usage of the English language.

Reply: The title “The Hijacking Exosomes-Mediated Cell Communication in Flavivirus Infection: New Insights on Virus-Host Interactions” was changed to “The Regulation of Flavivirus Infection by Hijacking Exosome-Mediated Cell-Cell Communication: New Insights on Virus-Host Interactions”.

3) All acronyms must be clearly defined for the reader the first time they appear. For example, on line 46, what does AB refer to?

Reply: The description of "ABs" as apoptotic bodies (Abs) was included in the first paragraph on line 67,

4) Line 48: fix: acceptor o target.

Repy: The “target” term was selected in this new submission (Line 72).

5) Line 57: fix: an essential factor to considerer.

Reply: The sentence was corrected: “How should the characterization of exosomes produced from the flavivirus-infected host and vector cells be studied?” (Line 84).

6) Line 68-70: as written the sentence that starts ‘While….’ Is not a complete sentence.  Please fix

Reply: The sentence was corrected: “Besides, techniques such as Western Blot, proteomic analysis, and next-generation sequencing (NGS) are used to study the composition of isolated exosomes” (Line 104).

7) Line 66: Cry-EM or Cryo EM as is used later in the MS.

Reply: The “Cryo-EM” was corrected (Line 100).

8) 2: I find this figure to be very complex and difficult to understand given the explanation in the legend.  My interpretation is that the authors are trying to include too much information into the graphic, making it difficult for the reader to digest what the take home points are.  Thus I do not find this figure to be particularly useful and would recommend that it be significantly re-drawn.

Reply: In the new version of the manuscript the information included in Figure 2 was included in Figures 2, 3, 4 and 5.

9) Line 184 – add the unit to the number in ….~100 (Exo-L) secreted…

Reply: The unit for the exosome size was included in this new submission (Line 197).

10) Section 3: This section is difficult for the non-expert reader to keep straight due to the inclusion of many details that are not put into proper perspective or flow.  Thus while it provides an overview of the literature in the area, it does not do it in a very approachable fashion – which I fear will significantly limit the impact of the review.

Reply: To facilitate the manuscript reading (and at the request of another reviewer), subheadings were added to sections 3 and 4 of the manuscript.

11) Line 326: the statement….EVs from ZIKV-infected mosquito cells can differentiate… implies that EVs differentiate – please reword for clarity.

Reply: The statement: “Regarding ZIKV, EVs from ZIKV-infected mosquito cells can differentiate and express CD11b+ on the surface of monocytes” was changed to “Regarding ZIKV, EVs from ZIKV-infected mosquito cells can induce differentiation and expression of CD11b+ on the surface of monocytes” in this new submission (Line 426).

12) Line 362: A hypothesis, not ‘an’ hypothesis.

Reply: The sentence: “5. Exosomes as a Transplacental Infection Route of ZIKV: An Hypothesis” was changed to “5. Exosomes as a Transplacental Infection Route of ZIKV: A Hypothesis” (Line 451).

Round 2

Reviewer 4 Report

The authors have responded adequately to the points raised in the original round of critiques.  I find the manuscript to be improved in terms of flow and overall clarity.